# Experimental Research of High-Quality Drilling Based on Ultrasonic Vibration-Assisted Machining

**DOI:** 10.3390/mi14081579

**Published:** 2023-08-10

**Authors:** Shuang Deng, Yu Guo, Songsong Lu

**Affiliations:** 1College of Mechanical and Electrical Engineering, Nanjing University of Aeronautics and Astronautics, Nanjing 210016, China; ds593384316@163.com; 2China Airborne Missile Academy, Luoyang 471009, China; lss009ly@163.com

**Keywords:** ultrasonic vibration-assisted drilling, micro-hole machining, motion trajectory, efficient machining

## Abstract

Micro-hole is widely used in various fields, and common machining methods of micro-hole in factories are drilling and electrical discharge machining (EDM), but because of low machining efficiency, these methods cannot meet requirements. Therefore, it is urgent to investigate a high efficient micro-hole machining technology to meet the demands of micro-hole machining in factory. Over past few decades, ultrasonic machining technology has developed rapidly and achieved good results in solving many critical machining problems in field of difficult-to-cut materials. Therefore, this paper builds an ultrasonic vibration-assisted drilling (UAD) experiments platform to combine micro-fine small hole drilling with ultrasonic machining technology for micro-hole multi-factor experimental research. Results show that UAD machining of micro-hole below diameter 0.5 mm is comparable to conventional drilling machining because of its high-frequency pulse intermittent cutting process, stable change in machining diameter, good stability of parameters such as shape tolerance roundness and cylindricity, small cutting force during cutting, small tool wear, and small surface roughness of inner wall of micro-hole. Compared with EDM, UAD has high efficiency and good stability of parameters such as diameter and roundness of shape tolerance. Comprehensive analysis of UAD can be used as an alternative technology solution for machining small holes in non-special requirements of metal materials in factory and has technical feasibility of stable batch production.

## 1. Introduction

Micro-hole, primarily small holes less than 0.5 mm in diameter [1], has been widely used in various fields, from precision instruments in aviation, aerospace, and military, to conventional equipment such as oil injection nozzles, molds, medical, and sanitary appliances. To achieve micro-hole machining, a lot of efforts have been made by researchers to improve machining methods in experiments, analytical modeling, and numerical simulation [2,3,4]. At present, more than 50 micro-hole machining methods and common small-hole machining methods are shown in Table 1 [5,6,7], which can be classified according to different principles of machining methods, and small-hole machining methods can be divided into two categories: traditional machining and non-traditional machining methods. Many of the above machining methods can be applied to micro small hole machining, such as drilling, EDM piercing, electrolytic machining, EDM-electrolytic compound machining, laser machining, ultrasonic machining, and electron beam machining.

Nowadays, micro-hole machining in hospitals is commonly used for drilling and EDM machining methods. Two machining methods are mainly shown as follows:(1)Drilling holes is one of most commonly used hole-machining methods in the field of machining, with its simple implementation, high machining efficiency, good dimensional accuracy, and low machining cost [8,9]. For holes with large diameters, small, deep diameters, and low accuracy requirements, drilling machining methods can generally meet machining requirements [10]. However, the drill bit is in a semi-closed state during the drilling process, which makes chip removal very difficult and prone to chip clogging. In addition, when the drill rotates fast, the machining gap is small, and the machining depth is large, the cooling medium does not easily enter the machining area, and it is difficult to cool and lubricate the drill, which increases the temperature of the drilling area and causes serious wear of drill, resulting in poor machining accuracy and easy damage to drill [11,12,13]. For hospital stainless steel, titanium alloy, and/or high hardness materials below 0.5 high-precision, small-hole machining is limited by drill strength, tool diameter, table drill speed requirements and/or conditions, and no effective drilling machining means to achieve rapid machining of micro-hole.(2)EDM perforation is the main machining method for machining large depth-to-diameter ratio deep small holes, which can process any conductive materials, such as titanium alloys, nickel-based high-temperature alloys, and/or difficult-to-machine materials, and has a wide range of applications in aerospace manufacturing [14,15,16,17]. EDM perforation mainly uses tubular electrodes that can perform high-pressure punching, and timely renewal and rapid discharge of galvanic corrosion products from the machining area, which enables efficient batch production of deep and small holes. A schematic diagram of EDM perforation is shown in Figure 1. However, in EDM deep small hole machining, the tool electrode cross-sectional area is small and under the influence of high-pressure punching fluid, irregular vibration is easily generated during the machining process; with an increase of hole machining depth, there are also problems such as heat dissipation, chip removal, and guiding difficulties, poor stability, high roughness, poor straightness, and low-dimensional accuracy [18,19,20,21]. At the same time, due to high-temperature melt removal material, hole walls often have re-microcracks, heat-affected zones, and recast layers [22], which cannot meet micro-hole quality requirements for high-temperature and high-pressure harsh environment work and require or means to improve machining surface quality and remove recast layers.

Ultrasonic machining is a machining method that uses ultrasonic vibrating tools to produce abrasive impact, polishing, hydraulic impact, and resulting cavitation in liquid media with abrasives or dry abrasives to remove materials, or to apply ultrasonic frequency vibrations to tools or workpieces in a certain direction for vibratory machining, or to use ultrasonic vibrations to bond workpieces to each other [23,24,25]. Over the past few decades, ultrasonic machining technology has developed rapidly, and there is more extensive research and there are more extensive applications in the field of ultrasonic vibration systems, deep and small hole machining, grinding, and polishing of drawing dyes and cavity molds, and ultrasonic composite machining, especially in the field of difficult-to-machine materials to solve many critical process problems and achieve good results. Therefore, there is an urgent need to conduct experimental research on the combined application of micro small hole drilling and ultrasonic machining technology to explore micro small hole ultrasonic drilling technology to achieve efficient machining of micro small holes.

## 2. Ultrasonic Cutting Technology

### 2.1. Ultrasonic Machining Principle

Ultrasonic wave is a frequency higher than 20 kHz, and in medium transmission of a sinusoidal energy wave, it has good directionality, energy that is easy to gather, high power, etc., ultrasonic machining using ultrasonic properties in metal medium transmission process to make metal plastic deformation and formation of best resonance point, so as to obtain a very large impact acceleration (10^4^~10^5^ times acceleration of gravity), so that tool to obtain following characteristics:(a)Kinetic energy: E = 1/2 mv^2^ (speed between 10^4^~10^5^ m/s);(b)Amplitude: 1~20 μm (small amplitude, will not affect machining accuracy);(c)Frequency: 20 kHz~50 kHz (High-frequency impact material, break material first, and remove it);(d)Cavitation: After contacting the tool, coolant forms a local negative pressure zone under the action of ultrasonic waves, causing rapid expansion of liquid interface and rupture, resulting in micro-excitation waves (5 × 10^7^ Pa high pressure, 4 KM jet stream).

The principle of ultrasonic machining is shown in Figure 2.

### 2.2. Principle of Ultrasonic Machining System

Ultrasonic cutting system converts electrical energy into ultrasonic energy through a transducer, using ultrasonic energy to make tool under the action of 20–50 KHz vibration frequency (i.e., 20,000–50,000 times per second) to obtain a very large impact acceleration (about 10^4^–10^5^ times acceleration of gravity), and main motion of machine tool compounded with high-frequency vibration along cutting direction, thus significantly reducing cutting resistance force and easily removing the residue of processed material. The principle of ultrasonic machining system is shown in Figure 3.

### 2.3. Ultrasonic Vibration-Assisted Drilling Motion Trajectory Model

When ordinary drilling is used, axial displacement of any point on cutting edge of tool is
(1)$$Z_{s}=-f_{v}nt$$

The vibration displacement function is given by
(2)$$Z_{D}=A\sin\left(2\pi ft\right)$$
where the negative sign indicates that tool is fed in the direction opposite to its positive direction; *f_v_* is feed rate; *n* is the spindle speed; *t* is cutting time; *Z_D_* is axial vibration displacement; *A* and *f* are ultrasonic amplitude and ultrasonic frequency, respectively.

From Equations (1) and (2), total axial displacement at any point on UAD tool is obtained as
(3)$$Z=Z_{D}+Z_{s}=-f_{v}nt+A\sin\left(2\pi ft\right)$$

To establish equations of UAD trajectory, assuming that tool radius is *r*, coordinate point of UAD at a certain moment is
(4)$$\left\{\begin{array}{l} X=r\cos\left(2\pi nt\right)\\ Y=r\sin\left(2\pi nt\right)\\ Z=-f_{v}nt+A\sin\left(2\pi ft\right) \end{array}\right.$$

Then, the angle of tool rotation at a certain moment is obtained by substituting the above equation to obtain trajectory of UAD at a certain angle of rotation.
(5)$$\left\{\begin{array}{l} X=r\cos\alpha \\ Y=r\sin\alpha \\ Z=-\frac{\alpha f_{v}}{2\pi }+A\sin(\frac{f\alpha }{n}) \end{array}\right.$$

## 3. Ultrasonic Experimental Platform

### 3.1. Experimental Systems

Ultrasonic cutting system adopts Super Kernen, which is shown in Figure 4. Its system has a function of automatic ultrasonic frequency tracking, when load changes cause resonant frequency to change, ultrasonic generator will automatically track change of frequency, so that tool always works on best resonant point. Ensure consistency of ultrasonic resonance during test process and ensure stability of test process data.

### 3.2. Experimental Equipment

Test equipment uses a three-axis milling center (Shenyang One Machine CNC Equipment Manufacturing Co., Ltd., Shenyang, China) and is rated at 12,000 rpm. The ultrasonic machining system accessories as shown in Figure 5, ultrasonic toolholder using CKN-XH (Huizhuan Technology Group Co., Ltd., Guangzhou, China) series ultrasonic toolholder, using wireless energy transfer, can adapt to common CNC milling teaching center machine structure, can realize free automatic tool change between ordinary tool holders and ultrasonic tool holders metal fully sealed housing, dustproof, and waterproof sealing effects and cutting fluid can Direct spraying, providing ultrasonic cutting process air bubble environment. Equipment auxiliary uses Kisler9257B (Kisler Corporation, Winterthur, Switzerland) cutting force measurement system to measure the axial force of the drill in the ultrasonic drilling process.

Ultrasonic generator using standard 30 kHz ultrasonic generator, ultrasonic frequency output 30~31 kHz adjustable, with output overload protection and short circuit protection, and positive power tracking function. In addition, the high energy conversion efficiency of ≥90%, and the whole system without overheating phenomenon ensure a long time of continuous, uninterrupted work to maintain stability of the test process.

## 4. Micro-Hole Machining Test Analysis

### 4.1. Experimental Parameters

In the machining test, GH4169 material is selected, and different size hole diameters (0.5 mm, 0.4 mm, 0.3 mm, and 0.2 mm) (depth 2.5 mm) are machined, and nine of each hole diameter are machined as a control. Tool (see Figure 6) and machine cutting parameters are shown in Table 2.

### 4.2. GH4169 Experimental Analysis of Plate Aperture Machining

Micro-hole machining results on GH4169 plate under ultrasonic vibration are shown in Figure 7.

#### 4.2.1. Ultrasonic Machining Results of 0.5 mm Micro-Hole on GH4169 Plate and Analysis

In Figure 8, it is shown that the results of 0.5 mm micro-hole machined on GH4169 sheet under ultrasonic vibration.

According to the results in Figure 8, the dimension of 0.5 mm micro-hole machining results on GH4169 plate under ultrasonic vibration are measured and shown in Table 3.

In addition, in Figure 9, it is shown that dimensions and roundness analysis of 0.5 mm micro-hole machining results on GH4169 plate are analyzed, and the dimensions of 0.5 mm holes on GH4169 plate under ultrasonic drilling are measured and the linear dimensional values of the holes are less than 0.51 mm, hole diameter variation is within 0.005 mm, shape tolerance roundness value <0.01, and roundness variation is stable within 0.005 mm.

#### 4.2.2. Ultrasonic Machining Results of 0.4 mm Micro-Hole on GH4169 Plate and Analysis

In Figure 10, it is shown that the results of 0.4 mm micro-hole machined on GH4169 sheet under ultrasonic vibration.

According to the results in Figure 10, the dimension of 0.4 mm micro-hole machining results on GH4169 plate under ultrasonic vibration are measured and shown in Table 4.

In addition, in Figure 11, it is shown that dimensions and roundness analysis of 0.4 mm micro-hole machining results on GH4169 plate are analyzed. From test results, we know that GH4169 plate using ultrasonic drilling 0.4 mm holes, dimensional tolerance value is small 0.408, hole diameter variation within 0.006, shape tolerance roundness value <0.007, roundness variation is stable within 0.005.

#### 4.2.3. Ultrasonic Machining Results of 0.3 mm Micro-Hole on GH4169 Plate and Analysis

In Figure 12, it is shown that the results of 0.3 mm micro-hole machined on GH4169 sheet under ultrasonic vibration.

According to the results in Figure 12, the dimension of 0.3 mm micro-hole machining results on GH4169 plate under ultrasonic vibration are measured and shown in Table 5.

In addition, in Figure 13, it is shown that dimensions and roundness analysis of 0.3 mm micro-hole machining results on GH4169 plate are analyzed. From test results, we know that GH4169 plate using ultrasonic drilling 0.3 mm holes, linear size value is small 0.308, hole diameter variation within 0.004, shape tolerance roundness value <0.007, roundness variation is stable within 0.005.

#### 4.2.4. Ultrasonic Machining Results of 0.2 mm Micro-Hole on GH4169 Plate and Analysis

In Figure 14, it is shown that the results of 0.4 mm micro-hole machined on GH4169 sheet under ultrasonic vibration.

According to the results in Figure 14, the dimension of 0.2 mm micro-hole machining results on GH4169 plate under ultrasonic vibration are measured and shown in Table 6.

In addition, in Figure 15, it is shown that dimensions and roundness analysis of 0.2 mm micro-hole machining results on GH4169 plate are analyzed. From test results, we know that GH4169 plate using ultrasonic drilling 0.2 mm holes, linear dimensional value is less than 0.21, hole diameter variation is within 0.007, shape tolerance roundness value <0.009, and roundness variation is stable within 0.005.

In summary, it can be seen that when using ultrasonic drilling 0.5 mm, 0.4 mm, 0.3 mm, and 0.2 mm holes, the hole diameter variation is within 0.005, the shape tolerance roundness value <0.01, the roundness variation is stable within 0.005, the UAD compared to CD machining, the hole diameter machining is stable, the variation is small, the inner hole shape tolerance roundness is smaller, the variation is small, and and the diameter size is better than when using CD machining.

### 4.3. Microporous Taper Analysis

The taper of hole is calculated based on entrance diameter and exit diameter of hole. As shown in Figure 16, the taper of hole in UAD is significantly smaller than the taper of a hole in conventional drilling. UAD can improve the stability of the drilling process. Compared with conventional drilling, the entrance diameter of hole in UAD is closer to theoretical diameter of hole, and taper of hole is smaller.

### 4.4. Cutting Force Analysis

Figure 17 shows the effect of machining parameters on average axial force. From Figure 17, it can be seen that axial force decreases with increase of spindle speed and increases with an increase of the feed rate. According to mechanical principle of metal cutting, as spindle speed increases or feed rate decreases, the thickness of uncut chips decreases, the cross-sectional area of chips also decreases, and the force required to cut chips decreases. When machining parameters are same, the axial force of UAD is less than that of CD. When the vibration frequency is the same, there is higher amplitude amd lower the axial force. At the amplitude of 4 μm, there is little difference in axial force at vibration frequencies of 22 kHz and 30 kHz. At the amplitude of 8 μm, axial force for 30 kHz vibratory drilling is less than that for 22 kHz vibratory drilling. The average axial force for UAD is reduced by 7.3% to 41.4% compared to CD.

### 4.5. Analysis of Cutting Burrs

The presence of burrs degrades the quality of parts, and their removal can significantly increase production costs. Figure 18 shows the effect of spindle speed and feed rate on average burr height. Average burr height decreases with increasing spindle speed and increases with increasing feed rate. The average burr height for UAD is significantly lower than that of conventional drilling.

### 4.6. Analysis of Surface Quality

Figure 19 shows the effect of spindle speed and feed rate on surface roughness of holes wall. As shown in Figure 19, surface roughness decreases with the increase of spindle speed and increases with increase of feed speed. As spindle speed increases or feed rate decreases, axial force decreases, plastic deformation of hole surface decreases, and surface roughness decreases. The surface roughness of hole walls produced by UAD is significantly lower than that of conventional drilling. Axial vibration of the drill bit during UAD polishes surface of holes wall, resulting in a smaller surface roughness.

The morphology of hole walls produced by UAD and conventional drilling was observed under a scanning electron microscope, as shown in Figure 20. It can be seen that the hole’s surface drilled with UAD is relatively smooth, with no obvious machining marks and only slight scratches. The machined surface of conventional drilling has spiral machining marks and delamination, and surface morphology has obvious consistency with surface roughness.

### 4.7. Tool Wear Analysis

Figure 21 shows tool wear comparison between UAD and conventional drilling. The machining parameters are spindle speed of 5000 r/min and feed rate of 12 mm/min. The vibration frequency during UAD was 30 kHz and amplitude was 8 μm. After drilling 30 consecutive holes, the wear of the drill bit was measured with an IFM G4 3D surface measuring instrument. The maximum cross-edge width of the drill bit was 22.9 μm in conventional drilling and 14.1 μm in UAD. A 38.4% reduction in maximum cross-edge width of the drill bit was achieved in UAD compared to conventional drilling. The maximum rear face wear width of conventional drilling bits was 36.4 μm, while maximum wear width of UAD bits was 21.7 μm, a reduction of 40.3% compared to conventional drilling. UAD can significantly reduce the amount of drill wear.

## 5. Experimental Validation Analysis

The verification test is conducted for a type of cooler core micro-hole machining, using UAD and EDM micro-hole machining for comparison and analysis. Test tool and cutting parameters are shown in Table 7.

EDM results are shown in Figure 22, and UAD results are shown in Figure 23.

UAD and EDM 0.1 micro-hole diameter size and shape tolerance roundness measurement results are shown in Table 8.

Test results analysis, UAD machining Φ0.1 micro-hole diameter and roundness is better than EDM machining. UAD is still a contact metal cutting process between tool and part material, and there is a plastic deformation process of metal in the micro-hole machining process of drill bit, and holes burr height condition is worse than EDM compared to EDM electrode discharge metal high-temperature melting process.

## 6. Conclusions

(1)Ultrasonic-assisted machining as a new special machining method, especially UAD in micro-hole machining has unique advantages, can solve the problem of micro-hole machining traditional drilling can not be processed or machining instability, while machining using general-purpose machine tools and tools to increase number of auxiliary ultrasonic machining system to achieve machining process, machining process of different hardness materials, good adaptability, especially in micro-hole machining of super-hard materials advantages are outstanding in micro-hole machining of difficult-to-cut materials.(2)After experimental analysis of UAD machining of micro-hole below 0.5 compared with traditional drilling machining, because of its machining process for high-frequency pulse intermittent cutting, machining diameter changes steadily, shape tolerance roundness and cylindricity, and/or parameters of good stability, cutting process cutting force is small, tool wear is small, micro-hole wall surface roughness is small.(3)Relative to EDM, UAD machining efficiency, machining diameter and shape tolerance roundness and or parameters of good stability, but UAD is still tool and part material contact metal cutting process, drill in the micro-hole machining process of the metal plastic deformation process, relative to EDM electrode discharge metal high-temperature melting process, hole burr height condition is worse than EDM machining.(4)A comprehensive analysis of UAD can be used as an alternative technical solution for the machining of small holes in non-special requirements of metal materials in hospitals, with stable batch production technical feasibility.

## Figures and Tables

**Figure 1 micromachines-14-01579-f001:**
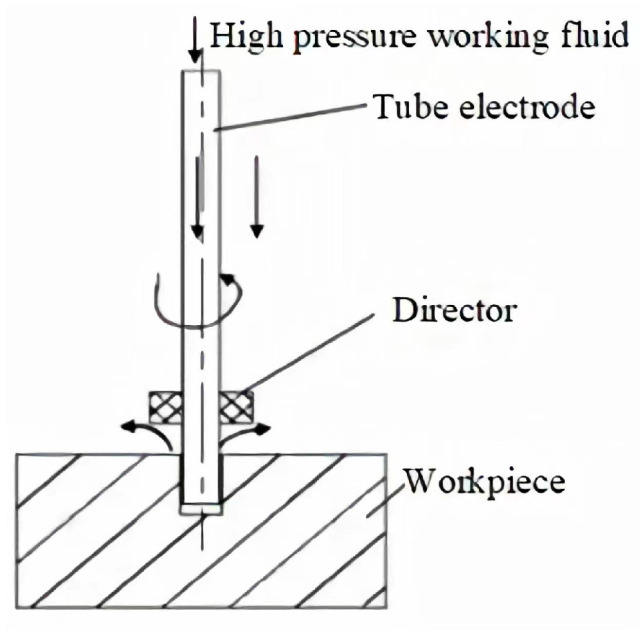
Schematic diagram of EDM piercing.

**Figure 2 micromachines-14-01579-f002:**
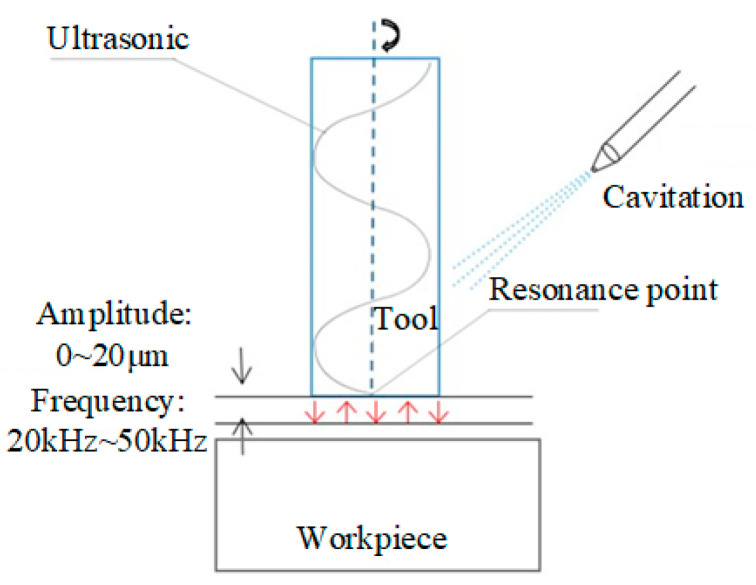
Schematic diagram of ultrasonic milling principle.

**Figure 3 micromachines-14-01579-f003:**
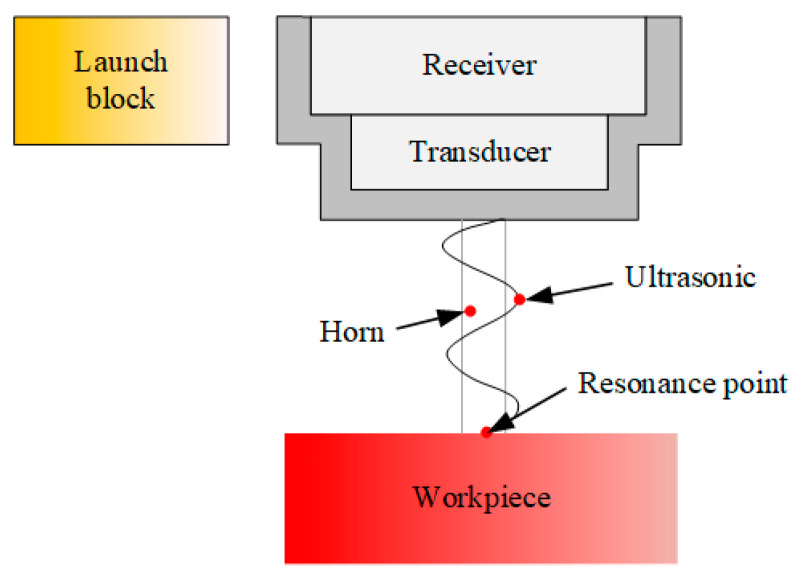
Schematic diagram of ultrasonic machining system.

**Figure 4 micromachines-14-01579-f004:**
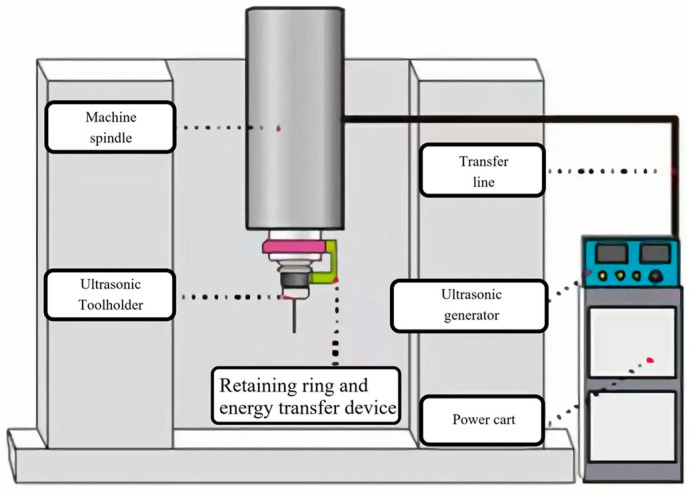
System view.

**Figure 5 micromachines-14-01579-f005:**
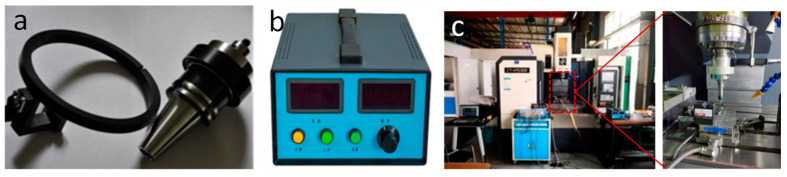
Ultrasonic milling system. (**a**) Ultrasonic shank and wireless energy transferring; (**b**) Ultrasonic generator; (**c**) Milling system.

**Figure 6 micromachines-14-01579-f006:**
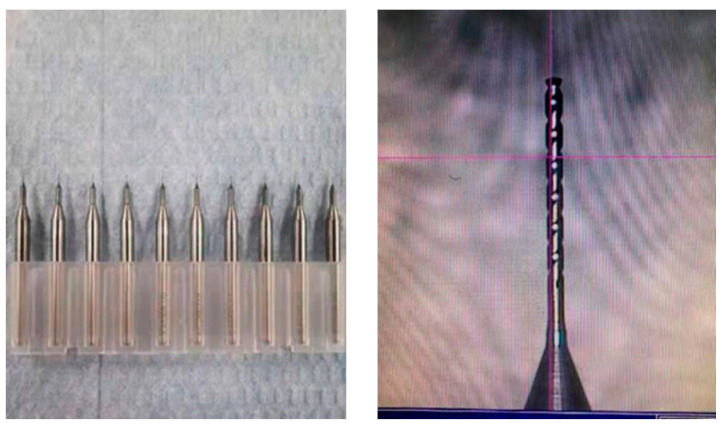
Test drill diagram.

**Figure 7 micromachines-14-01579-f007:**
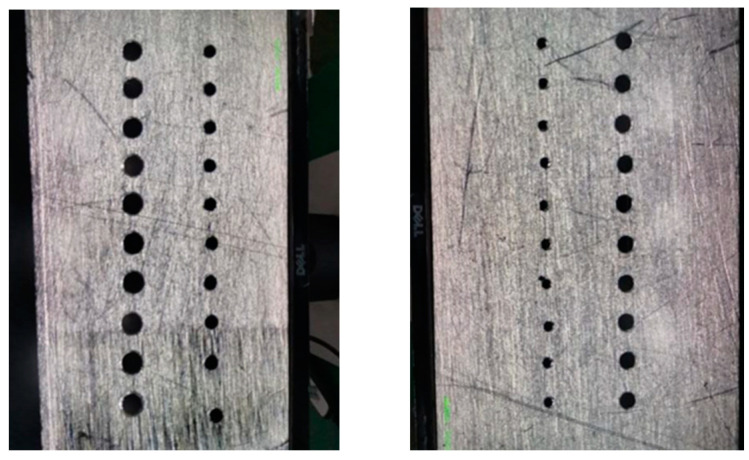
Ultrasonic micro-hole machining of GH4169 plate.

**Figure 8 micromachines-14-01579-f008:**
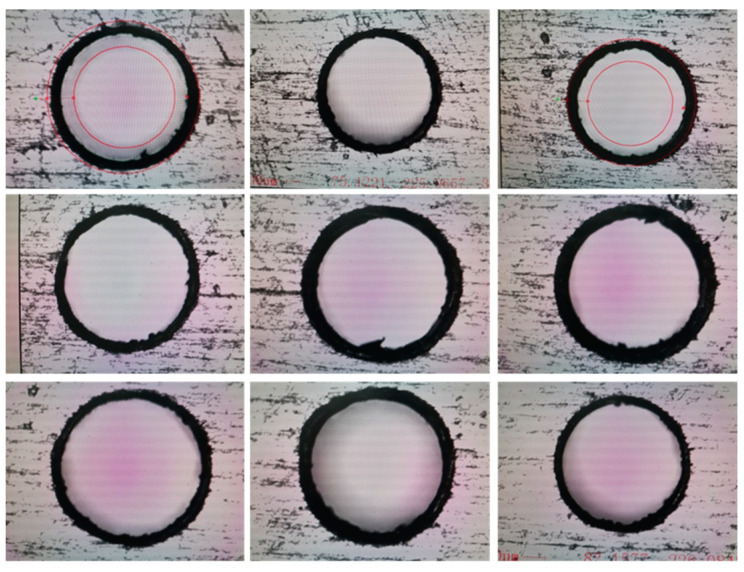
0.5 mm micro-hole machining results on GH4169 plate under ultrasonic vibration.

**Figure 9 micromachines-14-01579-f009:**
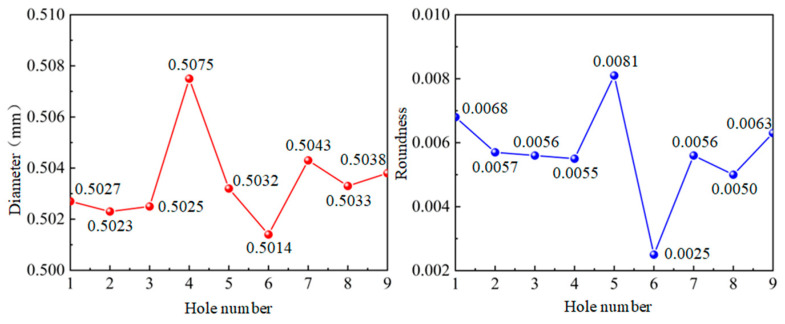
Dimensions and roundness analysis of 0.5 mm micro-hole machining results on GH4169 plate.

**Figure 10 micromachines-14-01579-f010:**
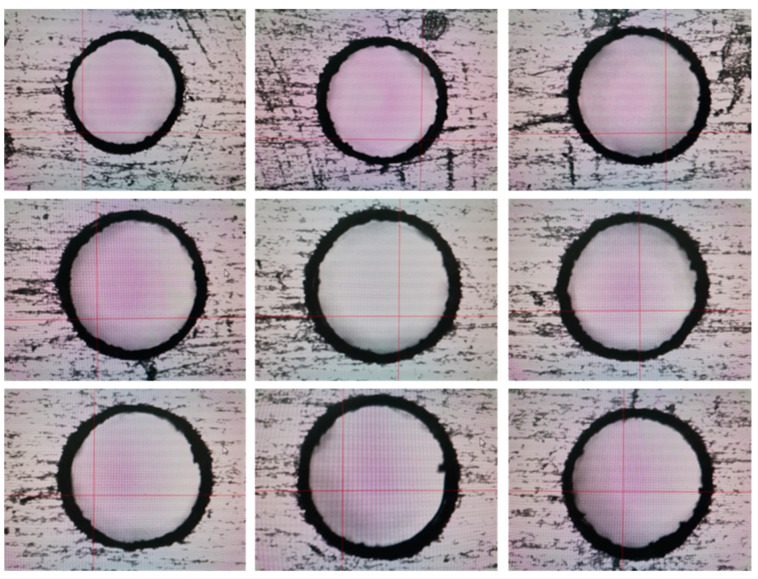
GH4169 plate ultrasonic 0.4 micro-hole machining results.

**Figure 11 micromachines-14-01579-f011:**
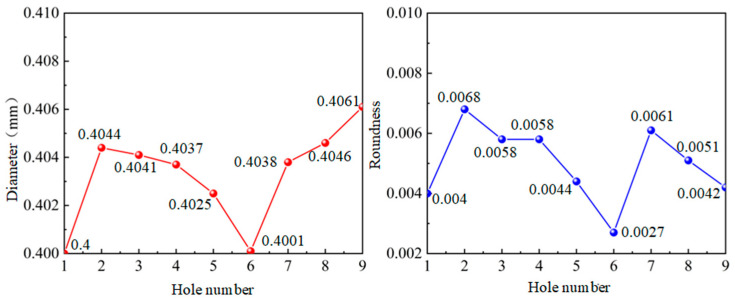
Dimensional and roundness analysis of ultrasonic 0.4 micro-hole machining of GH4169 plate.

**Figure 12 micromachines-14-01579-f012:**
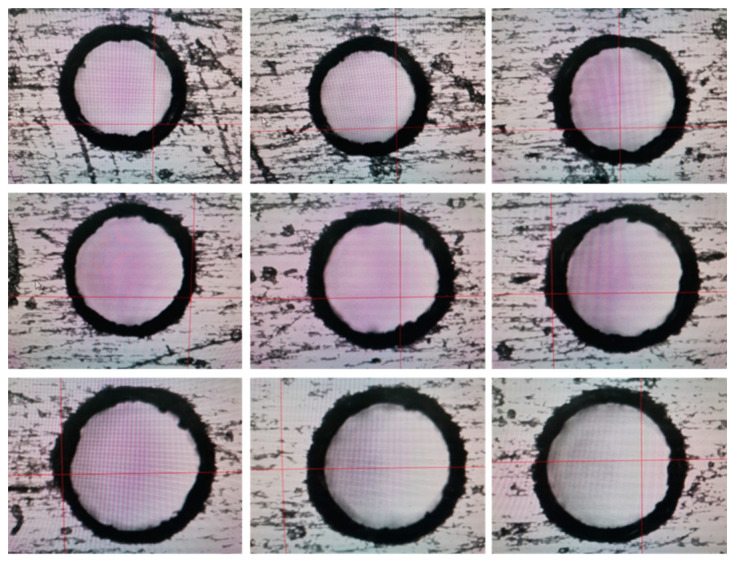
Results of ultrasonic 0.3 micro-hole machining of GH4169 plates.

**Figure 13 micromachines-14-01579-f013:**
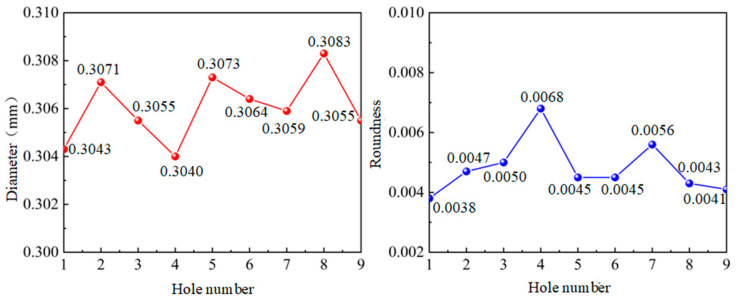
Analysis of size and roundness of ultrasonic 0.3 micro-hole machining of GH4169 sheet.

**Figure 14 micromachines-14-01579-f014:**
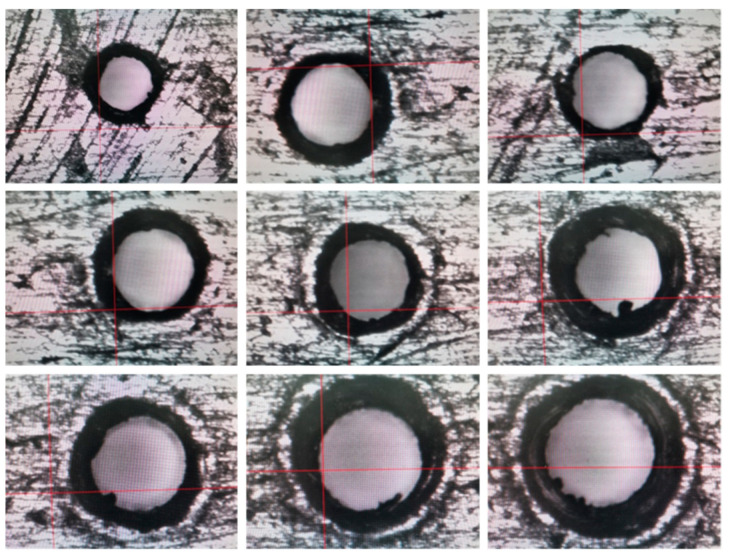
Results of ultrasonic 0.2 mm micro-hole machining of GH4169 plates.

**Figure 15 micromachines-14-01579-f015:**
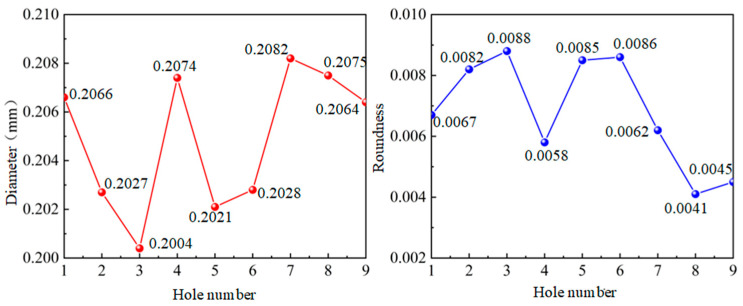
Dimensional and roundness analysis of ultrasonic 0.2 mm micro-hole machining of GH4169 plate.

**Figure 16 micromachines-14-01579-f016:**
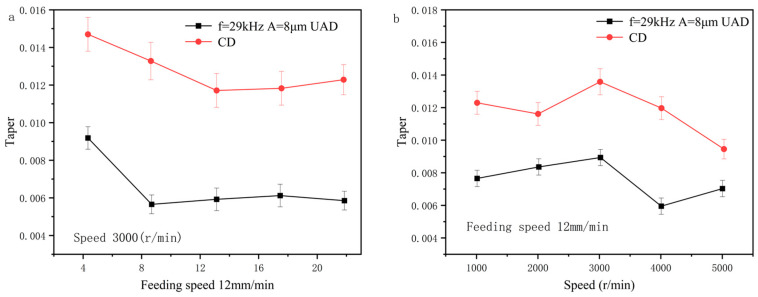
Effect of machining parameters on hole taper: (**a**) Effect of rotational speed on hole taper; (**b**) Effect of feed rate on hole taper.

**Figure 17 micromachines-14-01579-f017:**
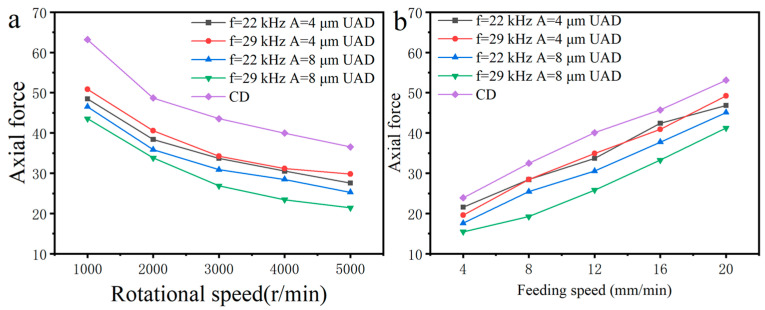
Effect of machining parameters on average axial force: (**a**) Effect of rotational speed on axial force; (**b**) Effect of feed rate on axial force.

**Figure 18 micromachines-14-01579-f018:**
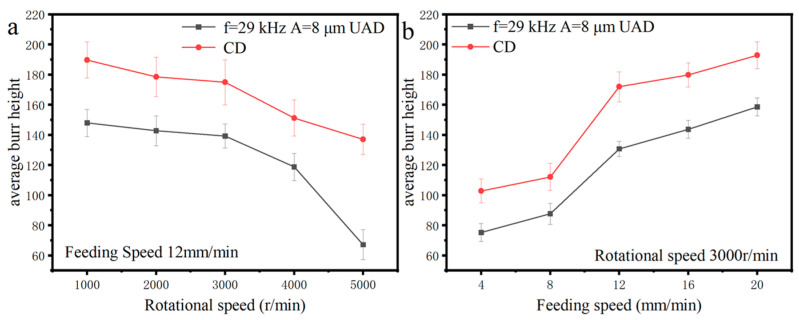
Effect of machining parameters on average axial force: (**a**) Effect of speed on average burr height; (**b**) Effect of feed rate on average burr height.

**Figure 19 micromachines-14-01579-f019:**
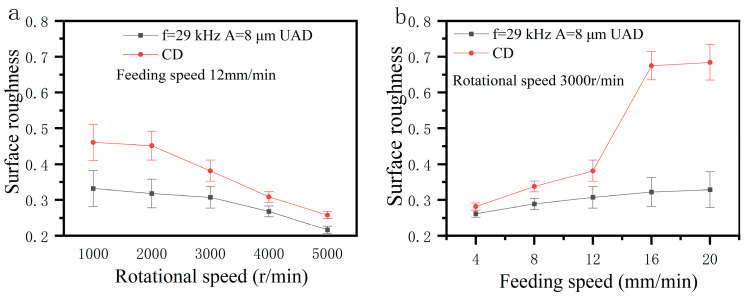
Effect of machining parameters on surface roughness: (**a**) Effect of rotational speed on surface roughness; (**b**) Effect of feed rate on surface roughness.

**Figure 20 micromachines-14-01579-f020:**
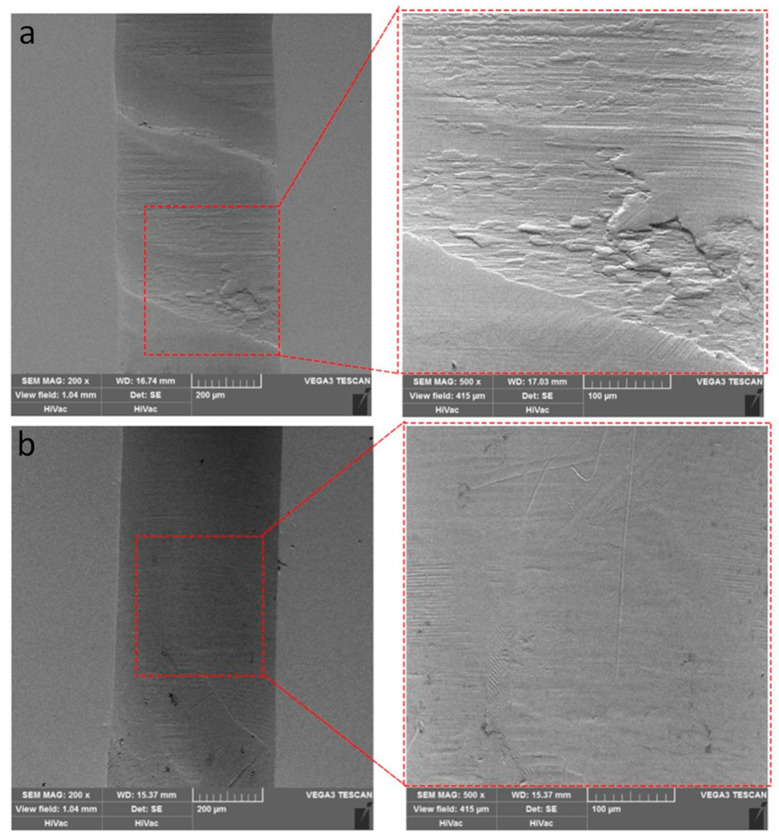
Hole wall shape of traditional drilling and UAD: (**a**) Hole wall shape in traditional drilling; (**b**) Hole wall shape in UAD.

**Figure 21 micromachines-14-01579-f021:**
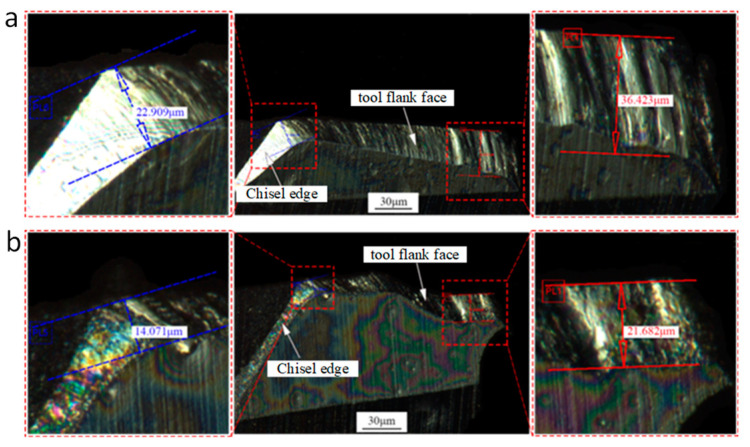
Comparison between conventional drilling and UAD bit wear: (**a**) Wear of conventional drilling bits; (**b**) Ultrasonic vibration assisted drilling bit wear.

**Figure 22 micromachines-14-01579-f022:**
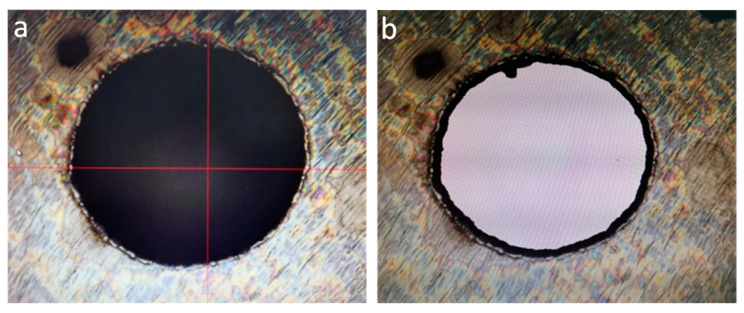
EDM results: (**a**) 0.1 micro-hole optical solid image; (**b**) 0.1 micro-hole optical swept surface imaging.

**Figure 23 micromachines-14-01579-f023:**
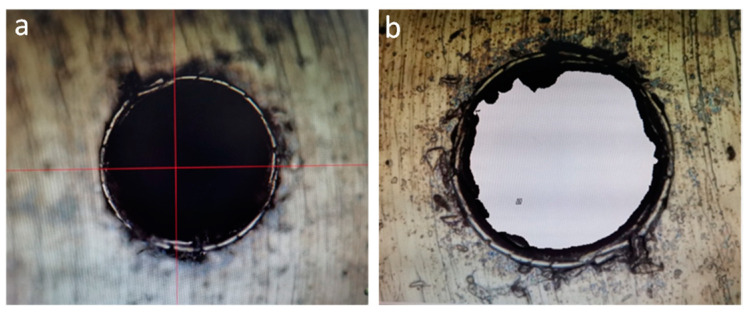
UAD machining results: (**a**) 0.1 micro-hole optical solid image; (**b**) 0.1 micro-hole optical swept surface image.

**Table 1 micromachines-14-01579-t001:** Common small hole machining methods.

Traditional Method	Non-Traditional Methods
Drilling, reaming, water jet machining, abrasive jet machining, punching, boring, end milling, honing, milling inner cavity, grinding, hinge	EDM perforation, electrolytic machining, chemical machining, electropolishing machining laser machining, photochemical machining, ion beam machining, ultrasonic machining

**Table 2 micromachines-14-01579-t002:** Cutting parameters of tools and machine tools.

Drill Diameter	Spindle Speed	Feed Speed
Φ0.2 mm, Φ0.3 mm, Φ0.4 mm, Φ0.5 mm	10,000 rpm	15–30 mm/min

**Table 3 micromachines-14-01579-t003:** GH4169 plate ultrasonic 0.5 micro-hole machining measured values.

Hole Serial Number	1#	2#	3#	4#	5#	6#	7#	8#	9#
Diameter (mm)	0.5027	0.5023	0.5025	0.5075	0.5032	0.5014	0.5043	0.5033	0.5038
Roundness	0.0068	0.0057	0.0056	0.0055	0.0081	0.0025	0.0056	0.0050	0.0063

**Table 4 micromachines-14-01579-t004:** GH4169 plate ultrasonic 0.4 micro-hole machining measured values.

Hole Serial Number	1#	2#	3#	4#	5#	6#	7#	8#	9#
Diameter (mm)	0.4000	0.4044	0.4041	0.4037	0.4025	0.4001	0.4038	0.4046	0.4061
Roundness	0.0040	0.0068	0.0058	0.0058	0.0044	0.0027	0.0061	0.0051	0.0042

**Table 5 micromachines-14-01579-t005:** GH4169 plate ultrasonic 0.3 micro-hole machining actual measured values.

Hole Serial Number	1#	2#	3#	4#	5#	6#	7#	8#	9#
Diameter (mm)	0.3043	0.3071	0.3055	0.3040	0.3073	0.3064	0.3059	0.3083	0.3055
Roundness	0.0038	0.0047	0.0050	0.0068	0.0045	0.0045	0.0056	0.0043	0.0041

**Table 6 micromachines-14-01579-t006:** GH4169 plate ultrasonic 0.2 micro-hole machining actual measured values.

Hole Serial Number	1#	2#	3#	4#	5#	6#	7#	8#	9#
Diameter (mm)	0.2066	0.2027	0.2004	0.2074	0.2021	0.2028	0.2082	0.2075	0.2064
Roundness	0.0067	0.0082	0.0088	0.0058	0.0085	0.0086	0.0062	0.0041	0.0045

**Table 7 micromachines-14-01579-t007:** Cutting parameters of tool and machine.

Ultrasonic Toolholder System	Ultrasound Frequency	Drill Diameter	Spindle Speed	Feeding Speed
CKN-XH	30 ± 1 kHZ	Φ0.1	10,000 rpm	15–30 mm/min

**Table 8 micromachines-14-01579-t008:** Test results of two machining methods.

Category	UAD	Drill Diameter
Diameter (mm)	1.006	1.012
Roundness (mm)	0.006	0.014
Burr height (mm)	0.035	0.011

## Data Availability

The data that support the findings of this study are available from the corresponding author upon reasonable request.

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
