# Peer review of "Experimental Research of High-Quality Drilling Based on Ultrasonic Vibration-Assisted Machining"

_micromachines, 2023, doi:10.3390/mi14081579_

Round 1

Reviewer 1 Report

1)      The Abstract and some other words in the text should not be bold.

2)      The abbreviation should be just shown when you first use it, while in the following statements there is no need to give the full explanation, such as UAD. But for EDM, you should give the full explanation.

3)      Figures like Fig. 1, 4, 17, 18, 19 are poor quality, high-definition figures should be replaced.

4)      Some expressions are not so right or suitable, revisions are needed such as, in Table2, “machine speed”(spindle speed), “feeding speed”(feed speed). And the unit of the diameter should be given.

5)      In “Author Contributions”, the surnames of authors should be given.

Minor editing of English language required

Author Response

On behalf of my co-authors, we would like to thank you for your constructive and helpful comments and the opportunity to revise the manuscript entitled “Experimental research of high quality drilling based on ultra-sonic vibration-assisted machining”(Manuscript ID: micromachines-2482126).

Those comments are all valuable and very helpful to improve our manuscript, and provide some significant guidance for our further research. We have studied comments carefully and revised the manuscript. And, we responded point by point to reviewer’s comments as listed below. In addition, the relevant changes are highlighted in red in the attached version. We hope these will make the manuscript more acceptable for publication.

The responses to the reviewer’s comments are as following:

Comment 1: The Abstract and some other words in the text should not be bold.

Response:

Thank you for pointing it out.

Your comment is very helpful to improve our manuscript. We admit that this is a very obvious mistake and we have revised the problems. The changes have been highlighted in red in the revised manuscript.

Comment 2: The abbreviation should be just shown when you first use it, while in the following statements there is no need to give the full explanation, such as UAD. But for EDM, you should give the full explanation.

Response:

Thank you for pointing it out.

Your comment is very helpful to improve our manuscript. The entire manuscript has been read thoroughly, and the problems of the abbreviation such EDM and UAD are revised in a uniform form. Then, the changes have been highlighted in red in the revised manuscript.

Comment 3: Figures like Fig. 1, 4, 17, 18, 19 are poor quality, high-definition figures should be replaced.

Response:

Thank you for pointing it out.

Your comment is very helpful to improve our papers’ preciseness. The high-definition figures (Fig. 1, 4, 16, 17, 18, 19) have been revised and replaced, and the changes have been highlighted in red in the revised manuscript.

Comment 4: Some expressions are not so right or suitable, revisions are needed such as, in Table2, “machine speed”(spindle speed), “feeding speed”(feed speed). And the unit of the diameter should be given.

Response:

Thank you for pointing it out.

Your comment is very helpful to improve our manuscript. The entire manuscript has been read and the relative expressions are checked thoroughly, and the changes have been highlighted in red in the revised manuscript.

Comment 5: In “Author Contributions”, the surnames of authors should be given.

Response:

Thank you for pointing it out.

Your comment is very helpful to improve our manuscript. We admit that this is a very obvious mistake. The surnames of authors have been added and we have made our utmost efforts to make the presentation as professional as possible. And, the changes have been highlighted in red in the revised manuscript.

To sum up, all questions have been answered.

We appreciate for Editor/Reviewer’ warm work earnestly, and hope that the correction will meet with approval. Once again, thank you very much for your comments and suggestions.

Sincerely yours,

Yu Guo

Reviewer 2 Report

1. The work focused on experimental investigation, but there are also other approaches such as analytical modeling and numerical simulation that should be introduced in introduction, for example:

  • Force prediction in ultrasonic vibration-assisted milling (https://doi.org/10.1080/10910344.2020.1815048)
  • A fundamental study of vibration assisted machining. (10.4028/www.scientific.net/AMR.264-265.1702)
  • Ultrasonic Vibration-Assisted Machining: principle, Design and Application (10.1007/s40436-015-0115-4)

2. The connection between section 2.3 and the experiments are unclear. Is the trajectory monitored in experiments to validate the model? Can surface roughness be predicted based on this model? You may refer to the surface roughness paper in comment #1.

3. Deviations of the experimental measurements should be provided with an error bar added in plots.

4. Only figure 17 shows the effect of different vibration frequency and amplitude. It would be good to have similar plots for surface roughness and tool wear.

5. Section 5 is only based on one machining condition in Table 7. How do you compare UAD and EDM in other conditions for more general applications?

6. There are other factors, such as machining temperature and residual stress, that are also very important to evaluate the machining process. Is there any measurement or analysis regarding these parameters?

Author Response

On behalf of my co-authors, we would like to thank you for your constructive and helpful comments and the opportunity to revise the manuscript entitled  “Experimental research of high quality drilling based on ultra-sonic vibration-assisted machining”(Manuscript ID: micromachines-2482126).

Those comments are all valuable and very helpful to improve our manuscript, and provide some significant guidance for our further research. We have studied comments carefully and revised the manuscript. And, we responded point by point to reviewer’s comments as listed below. In addition, the relevant changes are highlighted in red in the attached version. We hope these will make the manuscript more acceptable for publication.

The responses to the reviewer’s comments are as following:

Comment 1: The work focused on experimental investigation, but there are also other approaches such as analytical modeling and numerical simulation that should be introduced in introduction, for example:

Force prediction in ultrasonic vibration-assisted milling (https://doi.org/10.1080/10910344.2020.1815048)

A fundamental study of vibration assisted machining. (10.4028/www.scientific.net/AMR.264-265.1702)

Ultrasonic Vibration-Assisted Machining: principle, Design and Application (10.1007/s40436-015-0115-4)

Response:

Thank you for pointing it out.

Your comment is very helpful to improve our manuscript. We have added the relative papers in introduction, and the changes have been highlighted in red in the revised manuscript.

Comment 2: The connection between section 2.3 and the experiments are unclear. Is the trajectory monitored in experiments to validate the model? Can surface roughness be predicted based on this model? You may refer to the surface roughness paper in comment #1.

Response:

Thank you for your valuable suggestion.

Your comment is very helpful for improving our papers’ preciseness. We admit that this is a very obvious research points.

As we know, the motion trajectory of the tool tip is related to the surface quality of the part. In this paper, we want to explore the influence of ultrasonic vibration on the machining process. In theory, the motion trajectory of the tool tip under ultrasonic vibration is indeed different from that of ordinary machining, and the experimental contents can also verify this theoretical idea from different perspectives. In addition, surface roughness can be predicted considering the motion trajectory of the tool tip under ultrasonic vibration, but this paper only verifies that ultrasonic vibration has a significant impact on the quality of micro hole machining and improve the quality of micro hole machining by regulating ultrasonic parameters.

As per your suggestion, we have thoroughly study the surface roughness paper in comment #1, and the prediction method of parts surface roughness of processed under ultrasonic vibration is investigated as our future tasks.

  The entire manuscript has been read thoroughly, and the connection between section 2.3 and the experiments have been analyzed in paper. Then, the changes have been highlighted in red in the revised manuscript.

Comment 3: Deviations of the experimental measurements should be provided with an error bar added in plots.

Response:

Thank you for pointing it out.

Your comment is very helpful to improve our papers’ preciseness. We admit that your comment is correct and the deviations of the experimental measurements should be provided with an error bar added in plots. However, in Fig.9, Fig.11, Fig.13, Fig.15, we have analyzed the error in the paper to simplify the figures. For Fig.16, Fig.18, Fig.19, the error bar have added in plots, and for Fig.17, the effect of machining parameters on average axial force is shown, and it can be seen that axial force decreases with increase of spindle speed and increases with increase of feed rate. According to mechanical principle of metal cutting, as spindle speed increases or feed rate decreases, thickness of uncut chips decreases, cross-sectional area of chips also decreases, and force required to cut chips decreases. When machining parameters are same, axial force of UAD is less than that of CD. When vibration frequency is same, higher amplitude, lower the axial force. At amplitude of 4 μm, there is little difference in axial force at vibration frequencies of 22 kHz and 30 kHz. At amplitude of 8 μm, axial force for 30 kHz vibratory drilling is less than that for 22 kHz vibratory drilling. Average axial force for UAD is reduced by 7.3% to 41.4% compared to CD.

Comment 4: Only figure 17 shows the effect of different vibration frequency and amplitude. It would be good to have similar plots for surface roughness and tool wear.

Response:

Thank you for pointing it out.

Your comment is very helpful to improve our manuscript. In this paper, 60 sets of experiments are conducted to analyze the effects of vibration frequency and amplitude on machining process by cutting force. Then, one vibration frequency and amplitude are selected to investigate its effect on the surface roughness and tool wear during micro-hole machining. In the future, we will comprehensively analyze the influence of different ultrasonic frequencies and amplitude on the surface roughness and tool wear.

Comment 5: Section 5 is only based on one machining condition in Table 7. How do you compare UAD and EDM in other conditions for more general applications?

Response:

Thank you for pointing it out.

Your comment is very helpful to improve our manuscript. We admit that this is a very obvious research points. As the response to the comment 4, 60 sets of experiments are conducted to analyze the effects of vibration frequency and amplitude on machining process by cutting force. Then, one vibration frequency and amplitude are selected to investigate its effect on machining process in this paper, which contains the comparison of UAD and EDM in other conditions for more general applications

Comment 6: There are other factors, such as machining temperature and residual stress, that are also very important to evaluate the machining process. Is there any measurement or analysis regarding these parameters?

Response:

Thank you for your valuable suggestion.

Your comment is very helpful to improve our manuscript. In machining, cutting force and cutting temperature are common physical phenomena, which can generate residual stress. In our lab, we have cutting force measurement equipment, temperature measurement equipment, residual stress equipment, dynamic signal measurement equipment, acoustic emission et al. which can be used to measure the relevant signals in cutting process for evaluating the parts machining effect. However, in this paper, the method of ultrasonic vibration in machining micro-hole is very effective and verifies by the experiments. Finally, we promise that we will conduct in-depth research on cutting temperature and residual stress et al. in future research.

To sum up, all questions have been answered.

We appreciate for Editor/Reviewer’ warm work earnestly, and hope that the correction will meet with approval. Once again, thank you very much for your comments and suggestions.

Sincerely yours,

Yu Guo

Round 2

Reviewer 2 Report

Comments are addressed properly.